# Quantitative Effects of Temperature and Exposure Duration on the Occurrence and Repair of Indirect Chilling Injury in the Fall Armyworm *Spodoptera frugiperda*

**DOI:** 10.3390/insects14040356

**Published:** 2023-04-03

**Authors:** Yoshiaki Tanaka, Keiichiro Matsukura

**Affiliations:** 1Institute of Agrobiological Sciences, National Agriculture and Food Research Organization (NARO), Tsukuba 305-8634, Ibaraki, Japan; 2Institute for Plant Protection, National Agriculture and Food Research Organization (NARO), Tsukuba 305-8666, Ibaraki, Japan

**Keywords:** biological invasion, local adaptation, maize, overseas migration, overwinter

## Abstract

**Simple Summary:**

The fall armyworm (FAW) *Spodoptera frugiperda* (J.E. Smith) is a long-distance migratory insect pest, and the invaded range of its recent expansion includes regions colder than the tropical and subtropical regions in East Asia. Adaptation to freezing and/or chilling injury during winter is required for the successful invasion and subsequent localization of newly invaded species. The mechanisms of injury caused by low temperatures are classified into three types (freezing, cold shock, and indirect chilling), and our study shows that indirect chilling injury, which is caused by the long-term exposure of insects to moderately low temperatures (3 to 15 °C), is most important for the survival of *S. frugiperda* during the winter. Adult *S. frugiperd* were more tolerant to moderately low temperatures than the larvae and pupae, but survival decreased significantly when adult *S. frugiperd* were exposed to temperatures of 9 °C or lower. Survival was improved by short-term daily exposure to higher temperatures, indicating the existence of a repair process for indirect chilling injury in *S. frugiperd*. These findings on indirect chilling injury and the repair process will improve the estimation of the potential distribution of *S. frugiperd* in temperate and colder regions.

**Abstract:**

The fall armyworm (FAW) *Spodoptera frugiperda* is a long-distance migratory insect pest, and the invaded range of its recent expansion includes regions colder than the tropical and subtropical regions in East Asia. In order to understand the potential distribution of *S. frugiperd* in temperate and colder regions, we quantified the effects of temperature and exposure duration on the degree of indirect chilling injury caused to *S. frugiperd* under laboratory conditions. The adults were more tolerant to moderately low temperatures (3 to 15 °C) than the larvae and pupae. Survival decreased significantly when adult *S. frugiperd* were exposed to temperatures of 9 °C or lower. A time–temperature model suggested that indirect chilling injury began occurring at 15 °C. Survival was improved by short-term daily exposure to higher temperatures, indicating the existence of a repair mechanism for indirect chilling injury in *S. frugiperd*. The degree of repair depended on the temperature, but the relationship was not a simple direct proportion. These findings on indirect chilling injury and repair will improve the estimation of the potential distribution of *S. frugiperd* in temperate and colder regions.

## 1. Introduction

The fall armyworm (FAW) *Spodoptera frugiperda* is a long-distance migratory insect pest originally distributed in the subtropical to tropical regions of South, Central, and North America [1]. Since the first outbreaks of FAW in West and Central Africa in 2016 [2], its distribution has expanded over large parts of America, Africa, Asia, and Oceania in just a few years [3]. The invaded range of the recent expansion includes regions colder than tropical and subtropical regions in East Asia [4,5]. FAW was first reported in southern Japan in 2019 was been collected at several sites in the subarctic region in 2020 [6]. In its native range, the FAW is widely distributed across central and eastern North America; however, its overwintering area is restricted to the southern regions of the continent because of insufficient cold tolerance and a lack of diapausing ability [7]. These facts suggest that the ability to endure winter climate is a key factor for determining the potential overwintering areas in both native and non-native ranges.

After the worldwide expansion of the FAW, several studies have proposed the potential distribution range of this insect, e.g., [8,9,10,11]. Most of these works constructed species distribution models from worldwide records of the collection and occurrence of the FAW and estimated the probability of distribution of the FAW from climate data, such as temperature and precipitation for each region. These species distribution models are useful for understanding the risk of future invasions of FAWs into new areas but may need revision, particularly in evaluating the overwintering ability of the FAW in temperate and colder regions. For example, Wang et al. [10] determined the contribution of winter temperature in their distribution model with the “maximum entropy” approach based on records of past occurrences of FAWs, but the record included both overwintering sites and sites with only temporary occurrences. A prediction model proposed by Ramirez-Cabral et al. [8] using the “climate change experiment” approach considered the quantitative contribution of winter temperature to cold stress, but the parameter values related to cold stress were determined from fragmental survival data of FAWs under low temperatures. These rough evaluations of the overwintering ability of FAWs could cause non-negligible errors in the assessment of its potential distribution.

An insect’s viability under low temperatures is characterized as cold tolerance after exposure to low temperatures. According to Lee [12], the mechanisms of injury caused by low temperatures are classified into three types. The most severe injury (and lethal to most insects) is freezing, which causes cellular dehydration, a harmful shift in pH, and mechanical damage to the cell structure. The second type of injury is called direct chilling injury (also called cold shock). This injury occurs when insects are not frozen but are exposed to a low temperature after rapid cooling. The sudden decrease in body temperature causes phase transitions in membrane lipids and the subsequent loss of membrane function. The last type of injury is indirect chilling injury, which is caused by long-term insect exposure to moderately low temperatures. The possible mechanisms for this injury include membrane phase transition, protein denaturation, and loss of ion homeostasis [13,14].

The cold tolerance of the FAW is well characterized. In North America, mature larvae pupate in the soil, but they often emerge as adults when the soil temperature is enough to complete pupal development because the FAW does not have the ability to diapause [15]. According to a laboratory study, no individuals survived longer than 24 h after exposure to temperatures below –5 °C [16], although their freezing temperature is lower than –5 °C (from –12 to –8 °C) depending on developmental stage [17]. This suggests that chilling injuries rather than freezing injuries are the major lethal factors for FAWs overwintering in temperate and colder regions. The developmental stage most tolerant to low temperatures differs among the reports. Perkins [18] reported that the egg was the most tolerant, with a 30% survival rate at –10 °C, whereas other reports suggest that the pupae and/or mature larvae have a much higher survival rate than the eggs and young larvae under low temperatures [17,19]. These discordant results are caused by differences in the methods used to evaluate cold tolerance. These previous reports evaluated the degree of freezing injury and cold shock in FAWs by examining freezing temperature and survival in relation to rapid cold hardening but did not examine the effects of indirect chilling injury on long-term survival, which is a crucial factor for determining the invasion range of invertebrates from tropical and subtropical regions into temperate and colder regions [20,21,22]. Therefore, it is necessary to clarify the effects of indirect chilling injury caused by moderately low temperatures (approximately 0 to 10 °C) in FAWs to understand how the cold tolerance of this species might affect its range expansion to colder regions.

Indirect chilling injury can affect the ability of insects to overwinter, so the repair of these injuries under fluctuating thermal conditions is of interest for estimating cold tolerance adaptability. Nedvěd et al. [23] proposed a time–temperature model to explain the quantitative contribution of exposure temperature and duration to survival, and this model is applicable for describing the degree of chilling injuries in some insect pests [21,24]. Nedvěd et al. [23] also reported that daily interruptions of cold exposure with short intervals at higher temperatures reduced mortality in the springtail *Orchesella cincta* (Linnaeus, 1758). Similarly, an increase in survival rates has been reported in some other insects, such as the firebug *Pyrrhocoris apterus* (Linnaeus, 1758) and the beetle *Alphitobius diaperinus* (Panzer) [14]. In nature, daily temperatures are never constant, so it is necessary to consider the repair mechanisms of indirect chilling injury by temporary exposure to higher temperatures when evaluating the overwintering ability of FAWs.

In this study, we first compared the longevity of the different developmental stages of FAWs at several moderately low temperatures and then applied a time–temperature model to quantify the survival of adult FAWs under moderately low temperatures. Finally, we compared the survival of adults under different fluctuating thermal regimes to generalize the effects of short-term exposure to higher temperatures in mitigating indirect chilling injury in FAWs.

## 2. Materials and Methods

### 2.1. Insects Used

We used a laboratory-reared strain of *S. frugiperda*, originally collected from a forage maize field in Koshi, Kumamoto Prefecture, Japan (32°52′43″ N, 130°44′25″ E) in October 2018. Larvae were reared on an artificial diet (Insecta LFS; Nihon Nosan Kogyo Co. Ltd., Kanagawa, Japan) in a plastic cup (Biocup cube, 147 × 147 × 80 mm, 770 mL; Risupack, Gifu, Japan; approx. 30 larvae per cup). After pupation, pupae were moved to a new plastic cup and kept without food and water until eclosion. Pupae were sexed using the morphology of the abdominal tipis where necessary. Emerged adults were kept in a transparent autoclave bag (Labbag, 610 × 810 mm; Ieda Trading Corp., Tokyo, Japan) along with medical cotton impregnated with 10% brown sugar water. Egg masses laid in the plastic bags were moved to a new plastic cup with an artificial diet. All stages of the laboratory-reared strain were reared at 25 °C with a 16 h light-to-8 h dark photoperiod.

### 2.2. Comparison of Cold Tolerance among Stages and Temperatures

Survival rates of adults, pupae, and mature larvae were compared under constant low-temperature conditions in an incubator (MIR-154-PJ; Panasonic, Osaka, Japan). For pupae, females and males were examined separately to see differences in cold tolerance between the sexes. Each insect was kept in a small plastic cup (Cleancup, Risupack, 86-mm Φ × 40-mm H) and held at 15 °C for one week to acclimatize to the cold. Each cup included a piece of artificial diet as food for the larvae or medical cotton soaked in 10% brown sugar water as food for the adults. After one week of cold acclimation, the insects were exposed to constant temperatures ranging from 3 to 15 °C at intervals of 3 °C (i.e., 3, 6, 9, 12, and 15 °C). The numbers of insects surviving and developing to the next stage (i.e., larva to pupa and pupa to adult) were counted daily for 30 days. Adults that could not hold onto the wall or stand on the bottom of the plastic cup were regarded as dead. For pupae, it is difficult to determine whether they are alive or dead from their appearance; therefore, pupae that had lost resiliency when pinched with tweezers were regarded as dead. Larvae that shrank and developed black on their skin were considered to be dead. Larvae and pupae that developed to the next stage were regarded as having survived until the end of the cold treatment. For each cold temperature, 50 larvae and pupae and 22–31 adults were examined.

### 2.3. Effects of Temperature on Repair of Chilling Injury in Adults

We compared the survival rates of adults exposed to five fluctuating thermal regimes. Adults were individually kept in small plastic cups (volume, 120 mL) with a piece of medical cotton soaked in 10% brown sugar water and were cold-acclimated by exposure to 15 °C for one week. The cold-acclimated adults were exposed to the following five fluctuating thermal regimes programmed into the incubator: 3 °C for 24 h (3 °C-24 h, control), 3 °C-22 h/15 °C-2 h, 3 °C-22 h/17.5 °C-2 h, 3 °C-22 h/20 °C-2 h, and 3 °C-22 h/22.5 °C-2 h. The rates of temperature increase and decrease in the regimes were approximately 1 °C/min. The number of surviving adults was counted daily for 30 days. Adults that could not hold onto a wall or stand on the bottom of the plastic cup were regarded as dead. A total of 20 to 44 adults were supplied for each treatment.

As additional treatments, we also exposed the adults to the following fluctuating thermal regimes to see any effects of the duration of exposure to higher temperatures on survival rates: 3 °C-20 h/15 °C-4 h (*n* = 25) and 3 °C-20 h/17.5 °C-4 h (*n* = 40).

### 2.4. Fitting Survival Rates to a Time–Temperature Model

The effects of temperature and exposure duration on the survival of adult *S. frugiperda* were quantified by fitting the data to a time–temperature model originally proposed by Nedvěd et al. [22], as follows:$$S\left(t,T\right)=\frac{\exp\left[a+bt\left(T\;–\;c\right)\right]}{1+\;\exp\left[a+bt\left(T\;–\;c\right)\right]}$$
where *S*(*t*, *T*) is the survival rate (0 or 1) relative to that at the constant 15 °C treatment given by exposure duration (*t*, days) and temperature (*T*, °C); *a* and *b* are parameters to be estimated, and *c* is an upper-limit temperature (°C) at which chilling injury occurs. The optimal values of *a* and *b* were estimated with a least-squares method after determining the best values for the upper-limit temperature *c* from 0 to 20 °C, as described by Matsukura et al. (submitted). The parameter estimation was performed in Python 3.9.10 (accessed on 31 May 2022, https://www.python.org/downloads/release/python-3910/) using the library “scipy” ver. 1.7.3.

### 2.5. Statistical Analyses

The survival curves from among the adults were compared to female pupae, male pupae, and larvae at each constant low temperature using a log-rank test with Bonferroni correction. The survival curves of the adults at different constant low temperatures and those at different fluctuating thermal regimes were also compared using a log-rank test with Bonferroni correction. The survival curves for the treatment regimes 3 °C-22 h/15 °C-2 h and 3 °C-20 h/15 °C-4 h and those for 3 °C-22 h/17.5 °C-2 h and 3 °C-20 h/17.5 °C-4 h were compared using a log-rank test. All statistical analyses were performed in Python 3.9.10 using the library “lifelines” ver. 0.27.0.

## 3. Results

### 3.1. Survival of Adults, Pupae, and Larvae at Constant Low Temperatures

We observed significant differences in survival among the developmental stages at all five temperatures (log-rank test with Bonferroni correction, α = 0.01) (Figure 1). Adults survived better than the pupae and larvae at 3 and 6 °C: about 70% of the adults survived, whereas almost all larvae and pupae died on the fourth day at 3 °C, and fewer than 20% of larvae and pupae survived 8 days of exposure at 6 °C, whereas 48% of the adults survived for 15 days. There were no significant differences between the survival curves of adults and larvae at temperatures of 9, 12, and 15 °C. The survival rates of the pupae were lower than those of the adults and larvae at most temperature. No significant differences in survival curves between the female and male pupae were observed at any temperatures. Most pupae and larvae that were counted as survivors at day 30 had developed to the next stage (i.e., adults and pupae, respectively) during the cold treatment.

### 3.2. Survival of Adults at Different Constant Low Temperatures

Survival rates of adults differed significantly among exposure treatments (log-rank test with Bonferroni correction, α = 0.01) (Figure 2). The survival at 15 °C was significantly higher than at all other temperature treatments except for 12 °C, and about 50% of adults survived 30 days of exposure to 15 °C. Survival rates significantly decreased as exposure temperature decreased from 9 to 3 °C.

The time–temperature model of the survival data showed a gradual increase in *R*^2^ from 0.256 to 0.925 with an increase in *c* from 0 to 15.06 °C (Appendix A). *R*^2^ varied less than ±0.00001 at a range of between 14.97 and 15.06 °C for *c*, and the null hypothesis that the observed survival data did not deviate from the expected values calculated by the model was not rejected at 100% accuracy (*p* = 1.000) within this range of *c*. We, therefore, adopted a function with parameters of *a* = 4.055 and *b* = –0.036 and estimated when c = 15.00 °C as a representative model of the time–temperature survival curve for adult *S. frugiperda* (Figure 3).

### 3.3. Repair of Chilling Injury in Adults at Different Thermal Regimes

The survival rates of the adults exposed to 3 °C with 2 h of daily exposure to various higher temperatures from 15 to 25 °C were significantly different among the higher temperature exposures (log-rank test with Bonferroni correction, α = 0.01) (Figure 4). Daily exposure to 25 °C for 2 h resulted in the highest survival rate, followed by exposures to 22.5 and 20 °C. Daily exposure to 15 °C for 2 h did not improve survival when compared to exposure at constant 3 °C.

Exposure to 15 °C for 4 h per day significantly improved the survival of adults compared to 2 h daily exposure to 15 °C (*p* < 0.001, log-rank test), whereas 4 h daily exposure to 17.5 °C resulted in poorer survival in the adults than in 2 h daily exposure treatment at 17.5 °C (*p* = 0.008) (Figure 5).

## 4. Discussion

The winter climate in temperate and colder regions is an important factor restricting the distribution ranges of insects [25]. Adaptation to freezing and/or chilling injury is required for the successful invasion and subsequent localization of newly invaded species, particularly those originating from warm regions. This study demonstrated the synergistic effects of low temperature and exposure duration on the survival of FAW adults, which were the most tolerant to indirect chilling injury among all the developmental stages tested (Figure 1). The synergistic effect appeared at about 15 °C in the time–temperature model analysis (Figure 3), and a significant decrease in survival was observed when the adult FAWs were exposed to temperatures of 9 °C or lower (Figure 2). The classification of cold injury by Lee [12] indicates that the injury to the adult FAWs observed here is indirect chilling injury because FAWs do not freeze until temperatures fall below approximately –10 °C [26], and cold shock (i.e., sudden decrease of temperature) could not occur in our experimental design. Indirect chilling injury caused by moderately low temperatures through the entire overwintering season, rather than freezing injury and cold shock caused by severe low temperatures and unrealistic sudden decreases of temperature, respectively, should be considered for estimating the potential distribution and risk of FAWs in temperate and colder regions, as was reported for other invasive invertebrates [20,21,22].

Improvement of survival to 2 h of exposure to “repair temperatures” of 17.5 °C and higher temperatures revealed temperature dependence, but not in a simple, direct proportion (Figure 4). Improvement in survival was not significantly different at repair temperatures between 17.5 and 22.5 °C, and the curves for 20.0 and 22.5 °C were very similar. This suggests that the repair process for indirect chilling injury in FAWs consists of two physiological phases that are activated at different threshold temperatures. The results of the 4 h exposure treatments for repair support this hypothesis (Figure 5). Survival in the 3 °C-20 h/15 °C-4 h treatment was significantly higher than that following 2 h exposure to 15 °C, and the curve was almost the same as that from the treatment at 3 °C-22 h/17.5 °C-2 h. On the other hand, exposure to 17.5 °C for 4 h resulted in poorer survival than the treatment with only 2 h of exposure to the same temperature, although 4 h of exposure improved FAW survival when compared to the control.

The reason for the decrease in survival rate from 4 h of exposure to 17.5 °C is unclear, but it is clear that the degree of recovery from indirect chilling injury at repair temperatures of between 15 and 22.5 °C shares the same upper limit. Further repair of indirect chilling injury requires the activation of other repair mechanisms that are activated at higher temperatures, as observed in the 3 °C-22 h/25 °C-2 h treatment. Several studies have proposed the underlying mechanism of the repair process in insects, together with confirming this phenomenon. For example, Koštál et al. [14] reported that adult *Pyrrhocoris apterus* survived significantly longer in a −5 °C-22 h/20 °C-2 h temperature regime than in constant −5 °C conditions and found that potassium ion concentration in haemolymph was stable for a longer period in the fluctuating temperature regime than in the constant temperature condition, suggesting a recovery of ion pumping systems in the cell membrane by temporal exposure to high temperatures. The stable potassium ion in haemolymph was also reported in *Drosophila suzukii*, which prolonged its lifespan >three-fold when it experienced daily 4 h exposures (including temperature ramping period) to 25 °C during exposure to 3 °C [27]. Metabolic and phospholipid profiling in *D. meranogaster*, which was exposed to 20 °C for 2 h daily, improved survival probability at low temperatures between 2 and 5 °C, and a remarkable correspondence in the time-course of changes was found between the metabolic network and the phospholipids profiles, suggesting fast homeostatic regeneration during the warm intervals [28]. Assuming that the primary contributor to indirect chilling injury is physiological deterioration, such as oxidative stress, disruption of osmoregulation, changes in membrane phase, and loss of ion homeostasis, as was reported for other insects, e.g., [14,29,30], then the recovery of membrane fluidity due to temporary exposure to higher temperatures may relieve the stress. Further physiological studies are needed to clarify the physiological mechanisms of indirect chilling injury and its repair in FAWs.

## 5. Conclusions

The biology of insects at low temperatures has been of interest as an adaptation to nature. While the physiological mechanisms of freezing injury and tolerance to freezing have been well documented, those in nonfreezing injuries (i.e., chilling injury) are poorly understood despite their importance in adaptation to the moderately low temperatures associated with winter in temperate regions. The present study on the cold tolerance of FAWs (I) confirmed the synergistic effects of low temperatures and exposure durations on the degree of indirect chilling injury caused to the insects, and (II) we found a novel aspect in the adaptation of the insects to chilling injury, whereby recovery from chilling injury depends on exposure to high temperatures for repair. Further studies are required to clarify the physiological aspects of tolerance to indirect chilling injury in FAWs. This will contribute to a better understanding and the better management of insect pests invading temperate regions from warmer regions.

## Figures and Tables

**Figure 1 insects-14-00356-f001:**
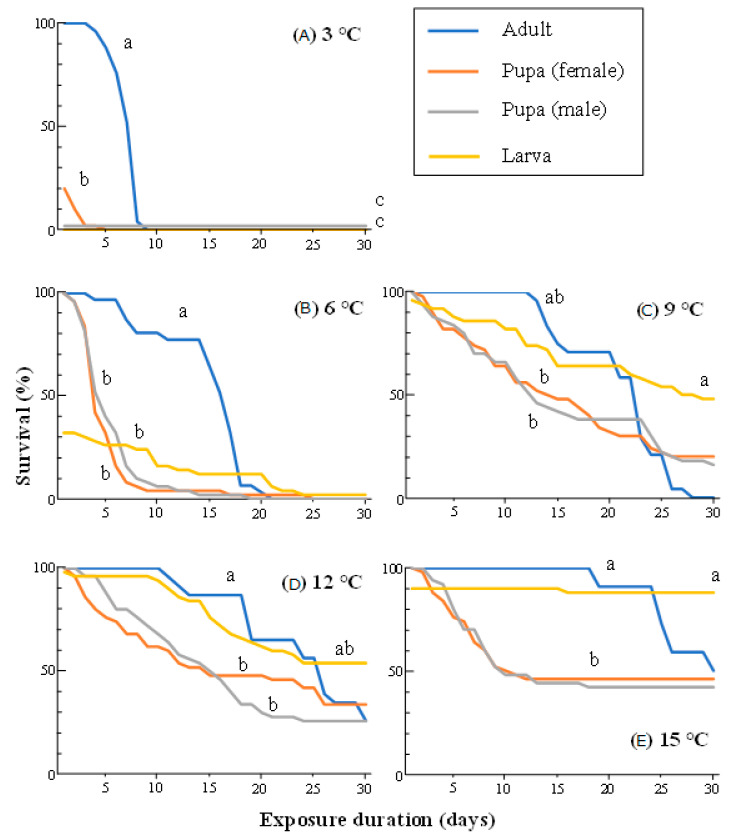
Survival curves for adults, pupae, and larvae of *S.frugiperda* exposed to constant low temperatures from 3 to 15 °C (**A**–**E**). IInsects at each developmental stage were individually kept in plastic cups and exposed to each temperature. Dead individuals were counted daily for 30 days. Each treatment used 22–31 adults and 50 pupae or larvae. Survival curves marked with the same letter (a–c) did not differ significantly within the same temperature exposure (log-rank test with Bonferroni correction, α = 0.01). Note that all larvae and pupae exposed to 3 °C died within 1 day after starting cold treatment.

**Figure 2 insects-14-00356-f002:**
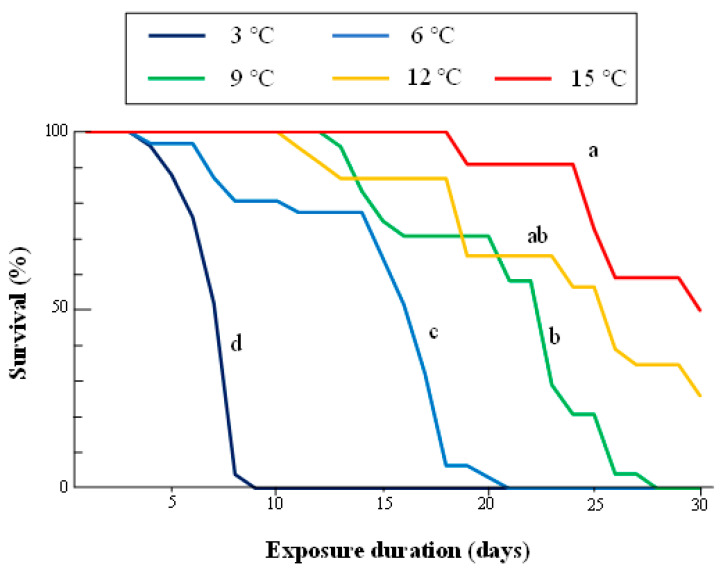
Survival curves of adult *S.frugiperda* at different constant low temperatures from 3 to 15 °C. See caption for Figure 1 details of the experimental design. Survival curves marked with the same letter (a–d) did not differ significantly within the same temperature exposure (log-rank test with Bonferroni correction, α = 0.01).

**Figure 3 insects-14-00356-f003:**
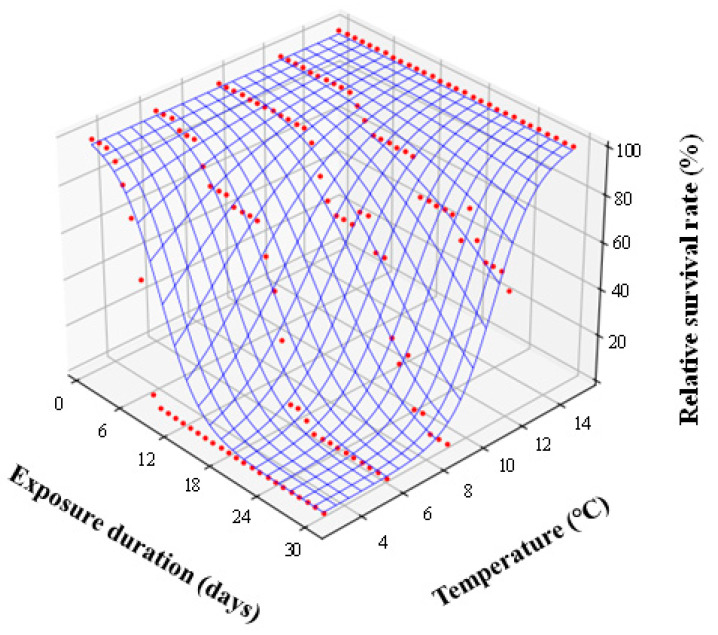
Effects of exposure time and temperature on the mortality of adult *S.frugiperda*. Red circles show the observed mortality under each set of conditions, and the blue curved surface indicates the expected mortality estimated by the best function with parameters of *a* = 4.055 and *b* = –0.036 under *c* = 15.00 °C in the time–temperature model proposed by Nedvěd et al. [23].

**Figure 4 insects-14-00356-f004:**
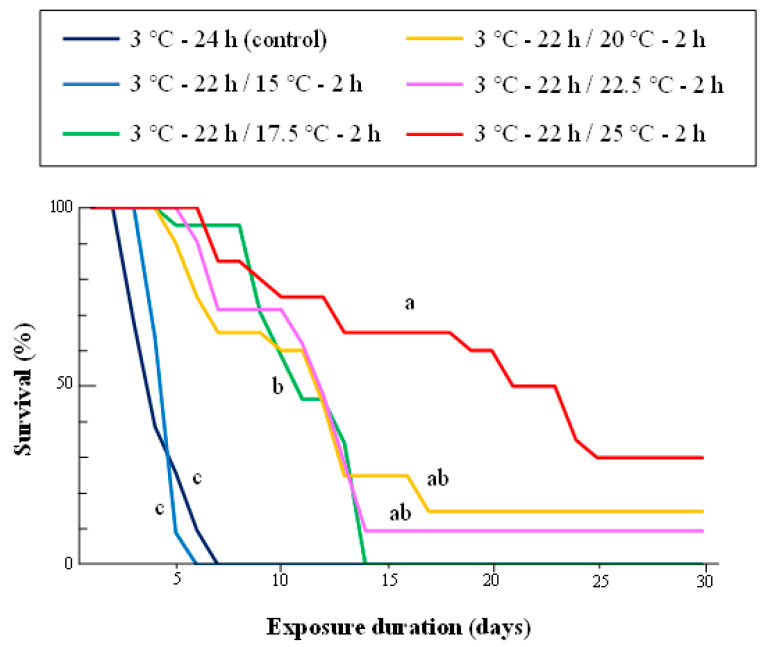
Survival curves of adult *S.frugiperda* under five fluctuating thermal regimes. See Figure 1 caption for details on the experimental design. Each treatment used 20–44 insects. Survival curves marked with the same letter (a–d) did not differ significantly within the same temperature exposure (log-rank test with Bonferroni correction, α = 0.01).

**Figure 5 insects-14-00356-f005:**
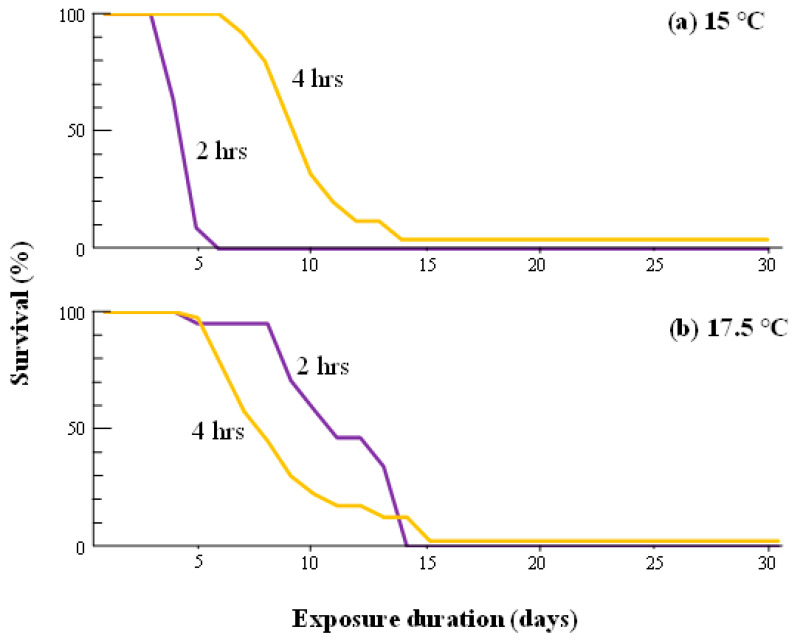
Survival curves of adult *S.frugiperda* exposed for 2 h or 4 h daily to temperatures of (**a**) 15 °C or (**b**) 17.5 °C, after exposure at 3 °C for the rest of the 24 h period. See Figure 1 caption for details on the experimental design. Each treatment used 25–44 insects. There were significant differences in survival between 2 h and 4 h of exposure at both 15 °C (log-rank test with Bonferroni correction, *p* < 0.001) and 17.5 °C (*p* = 0.008).

## Data Availability

The datasets used and analyzed during the current study are available from the corresponding author upon reasonable request.

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
