# Peer review of "Quantitative Effects of Temperature and Exposure Duration on the Occurrence and Repair of Indirect Chilling Injury in the Fall Armyworm Spodoptera frugiperda"

_insects, 2023, doi:10.3390/insects14040356_

Round 1

Reviewer 1 Report

The authors have detected the low tempetature adaptive of the fall armyworm by focusing on the indirect chilling injury and its reparing process. however, I have not seen the authors conclusion that the adults are more tolerant to moderate lower temeratures, which are the moderate low temperatures? most references have reported that the pupa is the most low temperature tolerant in this insect species. and why the surviuval rate curves of pupa during 3 centigrate exposure does not initiate from the 100% in fig1,also, why the larva curves were not appeared? Secondly, the data of the fig1 and fig2 for adutls survival curve is repetitive work. therefore, the authors should be clearly illustrate all the questions before the manuscript could be considerated to accept.

Reviewer 2 Report

This short but very informative manuscript represents the results of the experimental study on survival of different developmental stages of an important agricultural pest and global invader, the fall armyworm Spodoptera frugiperda under different constant and fluctuating thermal regimes to different extent causing chilling injury. The experiments were well designed and conducted. The statistical analysis is correct. The text is well and clearly written. The results of the study are not only important for fundamental insect ecology but also can be used for the estimation of the potential for the further invasion of the fall armyworm. Thus, the manuscript can be published almost “as it is” and only few improvements are required (see below).

Line 131: Please, explain why the data on male and female adults were pooled whereas the data on male and female pupae were analyzed separately.

Line 298: I think that for many readers the repair of indirect chilling injury will be quite a new phenomenon. Thus, I would suggest to describe the results of the cited studies [14, 26, 27, 28 and, if possible, some others] in more details: what insects were used, what combinations of temperatures and what exposure durations were tested, what hypotheses (if any) about underlying mechanisms were forwarded, etc. This kind of ‘micro-review’ (for one paragraph) would substantially increase the total value of the paper and, in addition, would highlight the novelty of your study.

Reviewer 3 Report

In this work, the authors examine the effect of fluctuating temperature regimes compared to constant temperature exposure on chilling injury and repair in S. frugiperda adults, pupae and larvae. They found significant differences in survival among different developmental stages with adults showing the most tolerance to constant cold exposure. However, adults maintained at specific low temperatures between 3 and 15 °C demonstrated chill injury based on high levels of mortality.

This manuscript is well written and experimental methods are clearly defined. Specific editorial suggestions are as follows:

 Abstract:

·       Avoid using abbreviations in abstract. Use S. frugiperda instead of FAW.

·       Add author name to species (i.e. Spodoptera frugiperda (J.E. Smith))(also add in start of introduction)

·       Lines15-16: make 2 sentences as “…indirect chilling). Our study shows that indirect chilling injury, which is caused by long-term…”

·       Line 18: define moderately low temperature

·       Line 33: change “simply” to “directly” if that is correct

·       Line 34: delete “the” and “process” to read “chilling injury and repair will…”

Introduction:

·       Line 43: be more specific regarding parts of America (i.e. south, central, north)

·       Lines 45-49: consider re-wording sentence to “Fall armyworm was first reported in southern Japan in 2019 and has been collected at several sites in the subarctic region in 2020. In its native range, FAW is widely distributed across central and eastern North America however, its overwintering area is restricted to southern regions of the continent because of ….”

·       Line 79: Describe what is known about the overwintering biology of these insects in nature.

·       Line 79: change “reported” to “characterized”

·       Line 84: add the word “cold” after most

·       Line 85: delete the word “some”

·       Line 87: change “under” to “to”; Change “these discordant” to “Inconsistencies in”

·       Line 97: consider changing “this ability” to “cold tolerance adaptability”

·       Lines 98-104: Add author names to species given in this section

·       Line 102: start sentence as “Similarly, an increase in survival rates has been reported in other insects…”

·       Line 105: add the word “mechanisms” after “repair”

Materials and Methods:

·       Lines 125, 135, and 151: I think you mean “sterile” instead of “sanitary”

·       Line 134: add cup volume

·       Line 143: change “judged to be dead” to “also considered to be dead”

·       Line 146: change “exposed” to “examined” and add comma after temperature

·       Line 158: Start sentence as “A total of 20 to 44…”

Discussion:

·       Line 265: change “clarified” to “demonstrated”

·       Lines 281-283: re-word this sentence. It is confusing as is.

Conclusions:

·       Change first sentence as follows: “The biology of insects….adaptation to various environmental conditions.”

·       Line 308: Change “has been poorly understood despite its importance…” to “are poorly understood despite their importance…” and change “of winter” to “associated with winter”

·       Line 310: Add the word “The” to start the sentence as “The present study…”

Figure captions:

·       Make caption format uniform, i.e. use either “Fig. X” or “Figure X”

·       Abbreviate Spodoptera frugiperda as S. frugiperda

·       Figure 3 – I am not sure you need to go out to so many decimal places for 15.00 °C
